# Evaluating seasonal hydrological extremes in mesoscale (pre-)Alpine basins at coarse 0.5° and fine hyperresolution

Joost Buitink[1], Remko Uijlenhoet[1], and Adriaan J. Teuling[1]

[1]Hydrology and Quantitative Water Management Group, Wageningen University, Wageningen, The Netherlands

**Correspondence:** Joost Buitink (joost.buitink@wur.nl)

**Abstract.** Hydrological models are being applied for impact assessment across a wide range of resolutions. In this study, we quantify the effect of model resolution on the simulated hydrological response in five mesoscale basin in the Swiss Alps using the distributed hydrological model Spatial Processes in Hydrology (SPHY). We introduce a new metric to compare a range of values resulting from a distributed model with a single value: the Density Weighted Distance (DWD). Model simulations are performed at two different spatial resolutions, matching common practices in hydrology: 500×500 m matching regional scale models, and 40×40 km matching global scale modeling. We investigate both the intra-basin response in seasonal streamflow and evapotranspiration from the high resolution model, as the difference induced by the two different spatial resolutions, with a focus on four seasonal extremes, selected based on temperature and precipitation. Results from the high resolution model show that the intra-basin response covers a surprisingly large range of anomalies, and show that it is not uncommon to have both extreme positive and negative flux anomalies occurring simultaneously within a catchment. The intra-basin response was grouped by land cover, where different dominant runoff generating processes are driving the differences between these groups. The low resolution model failed to capture the diverse and contrasting response from the high resolution model, since both the complex topography and land cover classes were not properly represented. DWD values show that, locally, the hydrological response simulated with a high resolution model can be a lot more extreme than a low resolution model might indicate, which has important implications for global or continental scale assessments carried out at coarse grids of 0.5×0.5° or 0.25×0.25° resolution.

## 1 Introduction

In current distributed hydrological modeling, we identify two approaches at opposite sides on the scale of application. On one hand studies are performed at global scale, and on the other hand studies are performed at regional or basin scales. The modeling approach generally affects the choice of spatial resolution, one of the key modeling decisions in hydrological modeling (Melsen et al., 2018). Most global studies are ran at rather coarse spatial resolutions (often at 0.5×0.5°) to investigate trends in the terrestrial water cycle as result of recent and projected changes in climate conditions (e.g. Luterbacher et al., 2004; Sánchez et al., 2004; Barnett et al., 2005; Beniston et al., 2007; Sheffield and Wood, 2008; Adam et al., 2009; Sheffield et al., 2012; Van Huijgevoort et al., 2014; Jacob et al., 2014). These studies often rely on standardized values such as the Standardized Precipitation Index (SPI) or Standardized Runoff Index (SRI) in order to quantify differences between different climatic regions

across the globe. Although recent global hydrological models are slowly shifting from relatively coarse resolutions to very fine resolution (hyperresolution, ~1×1 km), this is not yet the state of the art (Wood et al., 2011; Bierkens, 2015; Bierkens et al., 2015). It is known that global simulations at high resolution improve predictions at small local scales (Bierkens et al., 2015). However, these global studies are limited by a lack of input data at hyperresolution or a lack of computational power (Beven

and Cloke, 2012; Beven et al., 2015; Melsen et al., 2016b). As a result, most of the global studies are still performed at a relatively coarse resolution. Even when global modeling at hyperresolution becomes state of the art, question remains how we should deal with simulations at these fine spatial scales, since the models parameterizations are developed on a coarser scale (Clark et al., 2017; Peters-Lidard et al., 2017).

Another type of hydrological studies are those at basin or regional scales. These studies mostly use distributed hydrological

models to simulate the hydrological response under climate change or climatic extremes (e.g. Middelkoop et al., 2001; Hurkmans et al., 2009; Driessen et al., 2010; Hurkmans et al., 2010; Wong et al., 2011; Immerzeel et al., 2012). Typical resolutions for these studies are similar to the previously mentioned hyperresolution or even finer. Since these studies have a narrower spatial focus than the global simulations, high resolution data is often easier accessible and the computational power is less of a limiting factor. Since it is typically assumed that there is no important discrepancy between dynamics at the local scale and

those at larger scale, results are often not standardized.

Both global and regional studies focus on reaching similar goals, yet with different methodologies. So far, no study has investigated how these two methodologies connect and how the modeling approach affects the results. The effect of model resolution on the simulated response has been investigated by numerous studies, either for regional climate models or for hydrological models (e.g. Haddeland et al., 2002; Leung and Qian, 2003; Carpenter and Georgakakos, 2006; Gao et al., 2006;

Lucas-Picher et al., 2012; Pryor et al., 2012; Lobligeois et al., 2014; Kumar et al., 2016; Melsen et al., 2016a). The majority of these studies agree that an increased resolution leads to more realistic model results, as small-scale variability is better represented. However, no study has investigated how anomalies in the simulated hydrological response depend on the modeling approach, or what the distribution of these anomalies within complex basins looks like.

In this study, we aim to bridge the large-scale (climatological) and regional-scale (hydrological) approaches by quantifying

how the simulated hydrological response depends on spatial resolution, including within-basin complexity. Despite the large body of literature addressing the problem of scaling in hydrology (e.g. Klemeš, 1983; Dooge, 1986, 1988; Blöschl and Sivapalan, 1995; Feddes, 1995; Kalma et al., 1995; Bierkens et al., 2000; Beven, 2001; Blöschl, 2001; Sivapalan et al., 2004; McDonnell et al., 2007; Sposito, 2008), a limited number of tools to quantify this problem are proposed. Our study presents a new metric to quantify the difference between a range of values with a single value: the Density Weighted Distance (DWD).

We use the recently developed Spatial Processes in Hydrology (SPHY) model to simulate five basins in the Swiss Alps, a region which is know for large variations in land cover and elevation (Gurtz et al., 2003; Verbunt et al., 2003; Jolly et al., 2005; Schaefli et al., 2007; Zappa and Kan, 2007; Bavay et al., 2013; Speich et al., 2015). Each basin is simulated at two resolutions: a typical resolution for regional scale models (~500×500 m, also matching "hyperresolution"), and a typical resolution for global scale models (~40×40 km, matching a 0.5×0.5° pixel). Model results from both resolutions are compared and diffe-

rences are quantified using the DWD metric. Since many hydrological processes are nonlinear or depend on thresholds, we

expect that the modeling approach can greatly affect the model results. These nonlinearities and thresholds imply that a small change in input data or initial conditions can lead to relatively large changes in hydrological response. When scaling over homogeneous catchments, the resulting nonlinear behavior is typically preserved. However when scaled over heterogeneous catchments, the resulting hydrological behavior might not be trivial. For example, Blöschl et al. (2013) investigated the 2013
flood of the Danube river caused by extremely heavy precipitation. They found that the discharge peak could have been higher, since not all precipitation fell as rain. In parts of the catchment that were high enough for the temperature to stay below zero degrees Celsius, a fraction of precipitation fell as snow and did not directly contribute to the discharge. Teuling et al. (2013) showed that evaporation increased during droughts, based on data from several headwater catchments in Europe. This was explained by the lack of rainfall coinciding with reduced cloud cover and increasing net radiation, which out-weighted the
effect of lower soil moisture conditions. Jolly et al. (2005) studied how vegetation responded to the extreme summer of 2003 in the Swiss Alps. They found that vegetation response was not homogeneous, but showed different responses per elevation zone. Finally, catchments in the Swiss Alps are known to show complex behavior due to the non-trivial response of snow and glaciers to extreme events (Verbunt et al., 2003; Zappa and Kan, 2007; Van Tiel et al., 2018). These examples indicate the complexity of the hydrological response and the variability in time and space in these regions. Therefore, we hypothesize that
the spatial resolution will play an important role in the simulated response, since many hydrological processes during extremes are inherently nonlinear combined with the fact that most of the variability occurs at scales smaller than the spatial resolution of global hydrological models.

## 2   Methods, model and data

### 2.1   Basins

For this study, we selected five mesoscale basins in the Swiss Alps. Not only is the response of these basins relevant at regional scale, these basins also contribute considerable amounts to large rivers in Europe. For example, the discharge of the Rhine consisted for almost 40% of melt water from the Swiss Alps during the warm and dry summer of 2003 (Wolf et al., 1999; Stahl et al., 2016). While not all basins are tributaries to the Rhine, they nonetheless provide important insight into our understanding of the behavior of mountainous catchments. The basins for our study were selected based on size (roughly corresponding to
the $0.5°\times0.5°$ pixel size), elevation range, land cover, data availability, and minimal human influence (as the model simulates the basins without reservoirs). Figure 1 shows the locations and digital elevation models of all catchments. Please note that not always the entire river basin is chosen; see Table 1 for the names, station identifiers used by Swiss Federal Office of the Environment (FOEN) and other characteristics. Two basin categories can be distinguished: high-elevation catchments with glaciers (Reuss, Rhone and Inn) and lower-elevation catchments without glaciers (Emme and Thur). We will refer to those
basin categories as Alpine and pre-Alpine, respectively.

**Table 1.** Statistics for each catchment (FOEN, 2016).

| Main river basin | Reuss | Rhone | Inn | Emme | Thur |
|---|---|---|---|---|---|
| FOEN station ID | 2056 | 2346 | 2403 | 2155 | 2181 |
| Name outlet station | Seedorf | Brig | Cinuos-chel | Wiler | Halden |
| Outlet elevation [m a.s.l.] | 438 | 667 | 1680 | 458 | 456 |
| Surface area [km$^2$] | 832 | 913 | 736 | 940 | 1085 |
| Mean elevation [m a.s.l.] | 2010 | 2370 | 2467 | 860 | 910 |
| Glaciation [%] | 9.5 | 24.2 | 8.5 | 0.0 | 0.01 |
| Mean annual precipitation [mm] | 1729 | 1339 | 1501 | 1745 | 1084 |

## 2.2 Data

The model is forced with daily precipitation and temperature from MeteoSwiss (MeteoSwiss, 2013, 2016). All forcing data is provided at a resolution of approximately 2×2 km. We focus on the period from 1993 to 2014, and selected four seasons with unusual precipitation and/or temperature values (winter of 1995, spring of 2007, summer of 2003 and autumn of 2002,

see Section 3.1 for more details). Land cover data was obtained from WSL (2016) and grouped into four classes: forest, grass, glacier and other. The latter class combines all sparse vegetation types, bare soil and rocks. Discharge observations are obtained from FOEN (2016). Catchment elevation, delineation and stream network are derived from the digital elevation model of Jarvis et al. (2008).

## 2.3 Hydrological model

The Spatial Processes in Hydrology model (SPHY) was used to simulate each basin at both resolutions. SPHY is a spatially distributed conceptual hydrological model, including representations of rainfall-runoff, cryosphere, evapotranspiration and soil moisture processes, as well as their nonlinearities and thresholds (Terink et al., 2015). The model runs on a daily time step and a user-defined spatial resolution. Sub-grid variability is taken into account via cell fractions, but only for snow and glacier fractions. SPHY has been applied in several studies around the globe, yet the study area of most studies are situated in the

Himalayas (Lutz et al., 2013, 2014; Terink et al., 2015; Lutz et al., 2016; Hunink et al., 2017; Wijngaard et al., 2017; Terink et al., 2018). A schematic overview of the model concept is presented in Fig. 2. Based on the daily average temperature, SPHY determines whether precipitation will fall as snow or rain. The liquid precipitation will fall on the land surface, where part of the water can be directed to the river as surface runoff, depending on the volume of water already present in the rootzone. The remainder infiltrates into the rootzone, where it is subject to evapotranspiration based on the type of land cover. Water in the

rootzone can either percolate to the subzone, or is transported to the river network as lateral flow. From the subzone, water can either move upward into the rootzone as result of capillary rise, or can percolate to the groundwater layer. Water in the groundwater layer will contribute to the river discharge as baseflow. Solid precipitation is added to the snow storage, where melting of snow is diverted to the stream network as snow runoff. Finally, part of the grid cell can consist of glaciers. A fraction

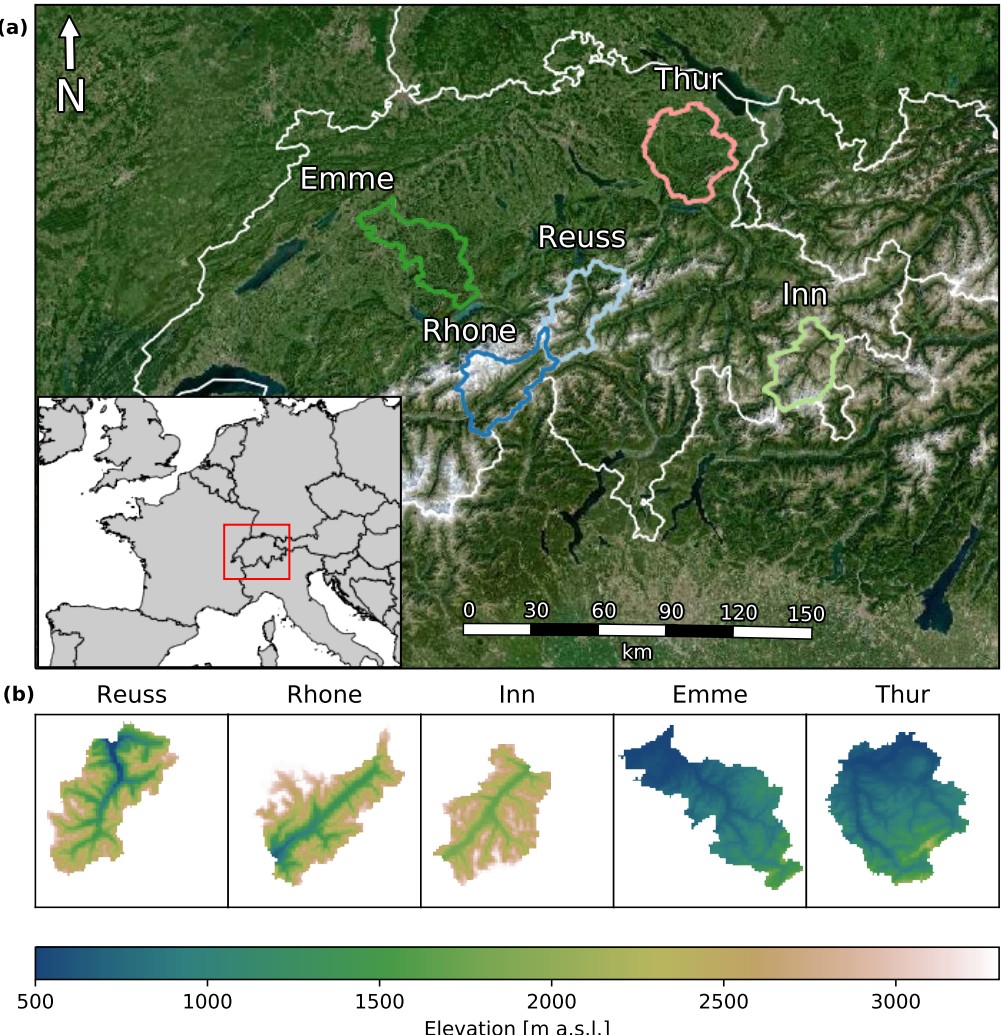

**Figure 1.** Overview of the location (a) and elevation (b) of the five basins used in this study. Names of the main river basin are plotted above the catchment border in (a). Each box in (b) corresponds to an area of ~40×40 km.

of the melted ice is added to the groundwater storage, and another fraction is transported to the river as glacier runoff. The glaciers in SPHY are fixed in space and time, so glaciers cannot extend and retreat. More information about the model structure and parameterizations are provided by Terink et al. (2015).

## 2.4 Model setup and calibration

SPHY was applied to each basin at two different resolutions: at ~500×500 m (corresponding to the regional scale resolution, and "hyperresolution"), and at ~40×40 km (corresponding to the global scale resolution of 0.5°×0.5°). This latter resolution

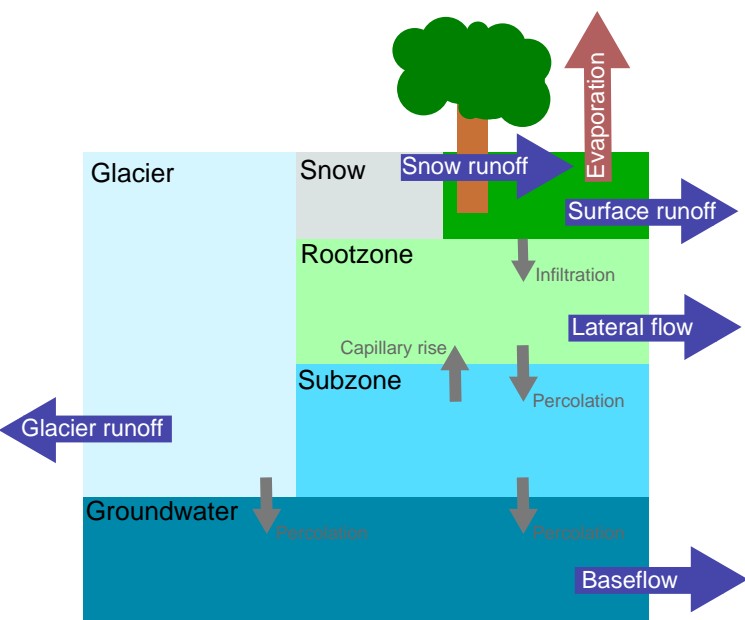

**Figure 2.** Schematic overview of the conceptualization in SPHY. Blue arrows represent fluxes contributing to total runoff generated in each model cell and small grey arrows represent fluxes between the different reservoirs. Overview is based on the more detailed concept by Terink et al. (2015).

implies that each basin was simulated as a single pixel. All input data were re-sampled to match the spatial resolution of the hydrological model. For the high resolution model we used bilinear interpolation to resample the forcing data for the high resolution model, and we averaged all cells within the 40×40 km pixel for the low resolution model. SPHY was calibrated individually for both resolutions and all basins using the L-BFGS-B algorithm (Zhu et al., 1997), by minimizing the sum

5 of squares of the residuals between monthly simulated and observed discharge. SPHY was calibrated over a period of five years (1997-2001), where the preceding year was used as spin-up period. These years were chosen to include both a relatively wet year (1999) and two relatively dry years (1997 and 1998). Four parameters were selected for calibration, all of which were found to influence the monthly discharge: rootzone depth, degree-day factor for snow melt, a parameter determining the fraction of water that can refreeze in the snow pack, and the critical temperature describing the point where precipitation falls

10 as snow. Since the L-BFGS-B algorithm is highly sensitive to the initial parameter guess, 10 different starting parameters sets were generated using Latin Hypercube Sampling to cover the parameter space (McKay et al., 1979). The calibration resulted in 10 new parameter sets per region and model type, and we selected the parameter set with the highest Kling-Gupta efficiency (Gupta et al., 2009). Using this parameter set, SPHY was ran from 1993 to 2014, where the first year was used as a spin-up period, resulting in 21 years of data used for analysis.

## 2.5 Anomalies and metrics

In this study, we only focus on the runoff and actual evaporation responses. We averaged all model output over three months, grouping the hydrological response per season: December, January and February for winter (DJF); March, April and May for spring (MAM); June, July and August for summer (JJA); September, October, November for autumn (SON). Standardized anomalies are used to quantify the magnitude of the deviation within each season, and are calculated for each individual model cell, using the following equation:

$$Z_{xi}^S = \frac{x_i^S - \mu_x^S}{\sigma_x^S},\tag{1}$$

where $\mu_x^S$ is the mean of variable $x$ in season $S$, $x_i^S$ is the value of variable $x$ for year $i$ in season $S$, $\sigma_x^S$ is the standard deviation of $x$ based on the same period, and $Z_{xi}^S$ is the dimensionless standardized anomaly of variable $x$ for year $i$ in season $S$. We note that most often climatologies are calculated based on time series of 30 years or more. We were not able to generate 30 years of data, because we only had sufficient data for the period 1993-2014. Since the focus of this paper is not on the absolute values, but on the patterns and relations, we do not expect different conclusions when longer time series would have been used.

Since the goal of this paper is to compare results from a high resolution model with results from a low resolution model, we require a suitable metric to quantitatively evaluate the difference between those results. Based on the previously discussed methodology, the high resolution model outputs a distribution of values, which needs to be evaluated against a single value from the low resolution model. Ideally, the metric provides robust information regardless of the shape of the distribution of the results from the high resolution model. A common option would be to calculate the percentile score of the low resolution model result within the high resolution model results. However, the percentile score does not provide information about the size of the error between the high and low resolution models. Another option would be the root mean square error (RMSE). The RMSE can be rewritten in terms of mean and variance, resulting in the following equation:

$$\mathrm{RMSE} = \sqrt{\sigma^2 + (\mu - Z_{\mathrm{low\_res}})^2},\tag{2}$$

where $\sigma^2$ and $\mu$ are the variance and mean of the (normalized) high resolution model results and $Z_{\mathrm{low\_res}}$ is the low resolution model result. However, when working with skewed or bimodal data (as visible in Fig. 10), the mean and variance are not sensible measures to describe the distribution of values.

Therefore we propose a new metric, which provides a measure of the distance between a single point and a distribution of values, regardless of the shape of the distribution. This metric does not only includes information on the difference in mean or median, but also on the width of the underlying distribution that the single value tries to represent. We call this new metric the Density Weighted Distance (DWD). DWD measures the distance between a single point and a range of values, weighted by the density of data that is present between the single point and the extent of the range of values. The extent is measured using the $5-95\%$ range to exclude the outliers, and the distances between the single point and the minimum and maximum extent

are multiplied by the percentile of data within this distance. DWD is defined as follows:

$$DWD = W_{\text{lower}} \cdot d_{\text{lower}} + W_{\text{lower}} \cdot d_{\text{lower}}, \tag{3}$$

$$W_{\text{lower}} = \max\left(0, \min\left(1, \frac{P_{\text{low\_res}} - P_{\text{lower}}}{P_{\text{upper}} - P_{\text{lower}}}\right)\right), \tag{4}$$

$$W_{\text{upper}} = \max\left(0, \min\left(1, \frac{P_{\text{upper}} - P_{\text{low\_res}}}{P_{\text{upper}} - P_{\text{lower}}}\right)\right), \tag{5}$$

$$d_{\text{lower}} = Z_{\text{low\_res}} - Z_{\text{high\_res}}^{5\%}, \tag{6}$$

$$d_{\text{upper}} = Z_{\text{high\_res}}^{95\%} - Z_{\text{low\_res}}, \tag{7}$$

where $W_{\text{lower}}$ and $W_{\text{upper}}$ are the weights used to weigh the distances $d_{\text{lower}}$ and $d_{\text{upper}}$. $P_{\text{low\_res}}$ is the percentile of $Z_{\text{low\_res}}$ within $Z_{\text{high\_res}}$. Both weights are corrected for the selected extent of the data ($P_{\text{lower}}$ and $P_{\text{upper}}$, default to 5% and 95%), and corrected between 0 and 1 if the $P_{\text{low\_res}}$ is outside the selected extent. The DWD concept is visualized in Fig. 3a. A property of this formulation is that high DWD values can mean two things: either that the low resolution model result is outside the range of values simulated with the high resolution model, or that the high resolution model results have high internal variability. This metric is aimed to measure the latter. We advise to always interpret DWD results together with the violin plots, to easier identify cases where the low resolution model result is outside the range of the results from the high resolution model .

The DWD can be interpreted as the difference in terms of number of standardized anomalies. DWD is zero when the high resolution data has zero variability, and when the difference with the low resolution model results is also zero. If the high resolution data has zero variability, but the result from the low resolution model is outside of this range, DWD will give the distance between the low and high resolution data, measured in number of standardized anomalies (see the "Flat" subplot in Fig. 3b.).

In order to illustrate the concept behind DWD and compare it to the previously mentioned metrics, Fig. 3b shows the different metrics using four synthetic example. The example with the "Flat" distribution assumes no variability in the high resolution model results. As a consequence, the violin plot is a horizontal line. Since there is no variability in the high resolution model results, RMSE and DWD give the same values. The percentile value is equal to zero, since the low resolution model result is outside of the high resolution model results. The three other examples in Fig. 3b illustrate that the percentile score does not give sufficient information to draw conclusions about the performance of the low resolution model, since they all received the same percentile score. The RMSE is able to catch the differences between the last two cases, but it does not accurately display the distance between the range of data from the high resolution model and the single point from the low resolution model. Furthermore, when working with skewed or bimodel data, the mean and variance are not the best indicators for the distribution of values. In contrast, DWD combines the spread of the high resolution results with the density of data points, resulting in a more sensible measure when dealing with skewed or bimodal data. We also compared the effect of selecting a different data range: $25 - 75\%$ instead of $5 - 95\%$. We conclude that this mostly influences results in terms of absolute size, but does not alter the relative differences much. We expect that when using the $25 - 75\%$ range, low resolution model results will be more often outside of this range than when using the $5 - 59\%$ range. Furthermore, we assume that all grid cells in the high resolution

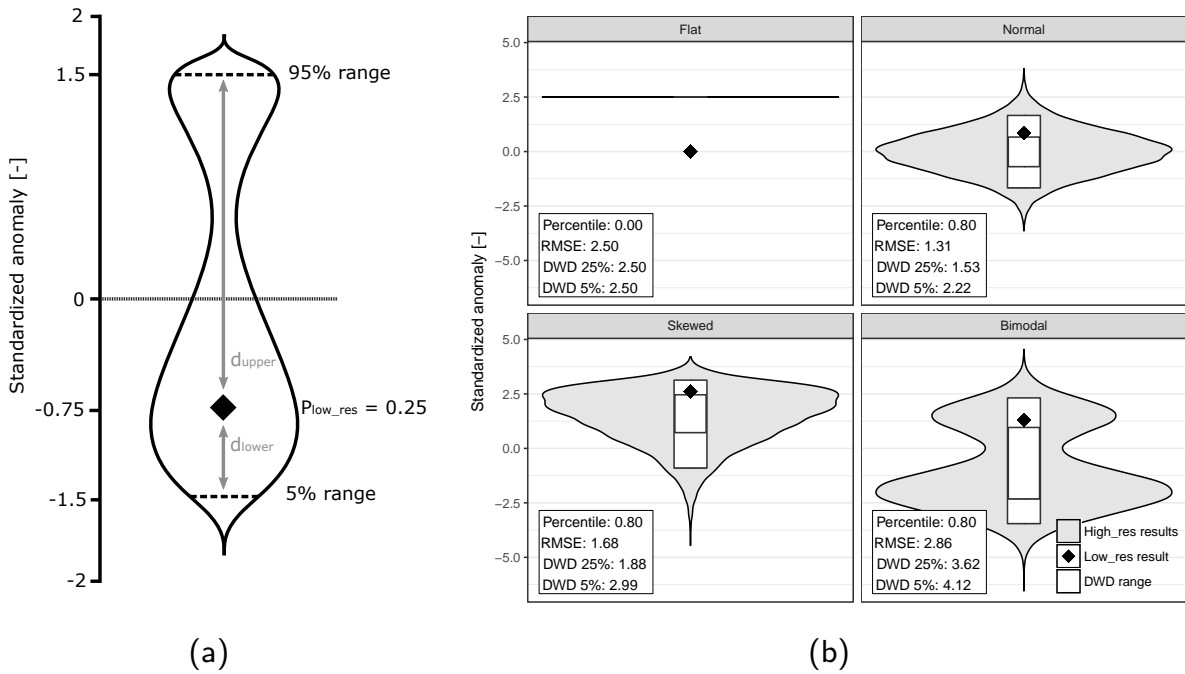

**Figure 3.** The concept behind Data Weighted Distance (a) and comparison between different metrics (b). Substituting the values from (a) into Eq. (3) gives the following result: $DWD \approx 1.92$. In (b), the violin plots represent the distribution of the high resolution model results and the diamond the single low resolution data point. The large box in (b) represents the $5 - 95\%$ data range, and the smallest box the $25 - 75\%$ data range.

model are equally important and will therefore use the largest data range to calculate the DWD, only excluding the outer 10% to remove any undesired behavior resulting from outliers.

# 3 Results and discussion

## 3.1 High resolution simulations

5    The key focus of this work is the catchment response to extreme seasons. To identify those extreme seasons, standardized precipitation and temperature anomalies are calculated for each season and basin (see Fig. 4). Since patterns are similar across the two catchment types, results of only two basins are shown in this figure. It should be noted that due to averaging values over three months, it is very likely that extreme events with a shorter duration are averaged out in this three-monthly time step.

   The highlighted dots in Fig. 4 show the extreme seasons selected for this study, for which the hydrological response is

10    analyzed. The seasons were selected based on unusual precipitation and/or temperature values: winter of 1994/95, spring of 2007, summer of 2003, autumn of 2002. Brönnimann et al. (2007) and MeteoSwiss (2017) both mention the high temperature during the spring of 2007 in Switzerland. The extremely warm and dry summer of 2003 is known to be the most extreme

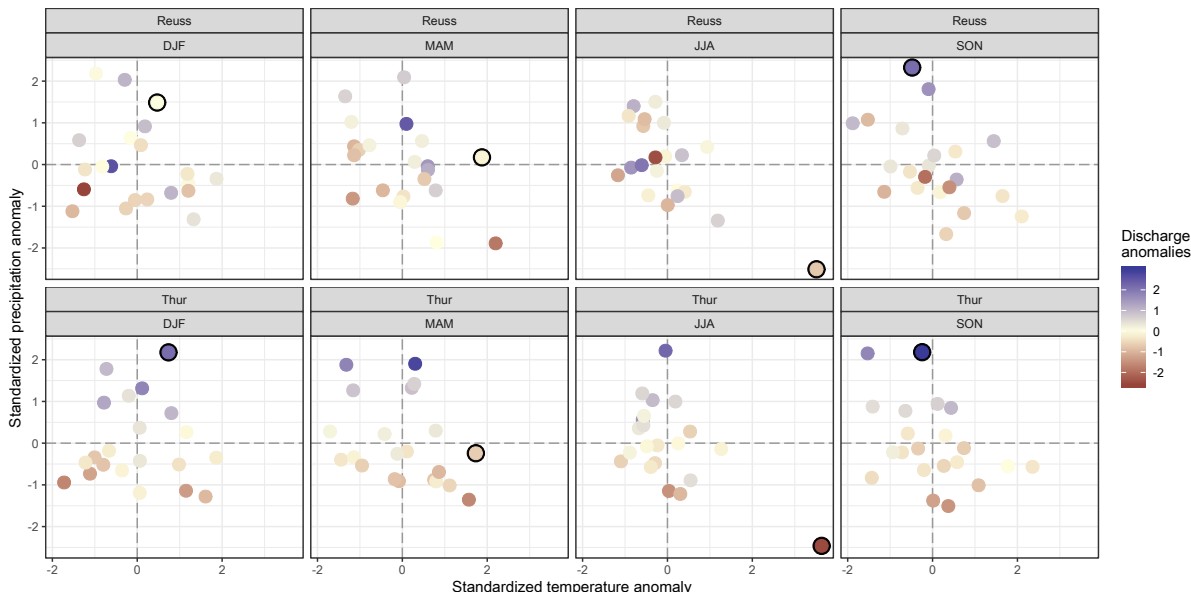

**Figure 4.** Relation between climate anomalies and observed discharge anomalies. Each dot represents a single season and is colored with the corresponding standardized observed discharge anomaly. Dots with black outline represent the selected extreme seasons (winter of 1995, spring of 2007, summer of 2003 and autumn of 2002).

summer in at least the last 500 years (Luterbacher et al., 2004; Zappa and Kan, 2007; Seneviratne et al., 2012). The extremely heavy precipitation during November 2002 caused mudflows in eastern Switzerland (Schmidli and Frei, 2005). No literature reference was found for the unusually wet winter of 1994/95.

The colors of the circles indicate the discharge anomalies. Discharge anomalies in the pre-Alpine basin seem to follow a distinct pattern, where high precipitation values often coincide with high positive discharge anomalies, and vice versa. Temperature also seems to influence discharge anomalies in the pre-Alpine basin, but this relation is less evident. The Alpine basin shows a much more random pattern, without any clear relation between temperature and/or precipitation. This indicates that the runoff generating processes are not consistently driven by either precipitation or temperature, but by a combination of both.

The calibration results for each basin are presented in Fig. 5a. This figure shows high Kling-Gupta efficiencies for all basins, indicating good model performance. In all basins, the high resolution model shows higher KGE values than the low resolution model, yet the values for the low resolution model still show relatively good performance. Only the winter discharge in the Alpine basins is underestimated by the model, at both resolutions. Discharge observations show an almost constant outflow during winter, which is most likely the result of human interference (reservoirs) (Fatichi et al., 2015). SPHY is not able to simulate this constant outflow and simulates discharge values close to zero. As a means of validating the model, we presented the spread (monthly standard deviation) around the monthly average discharge in Fig. 5b, excluding the years used for calibration. The high resolution model again shows better values than the low resolution model, and the spread around

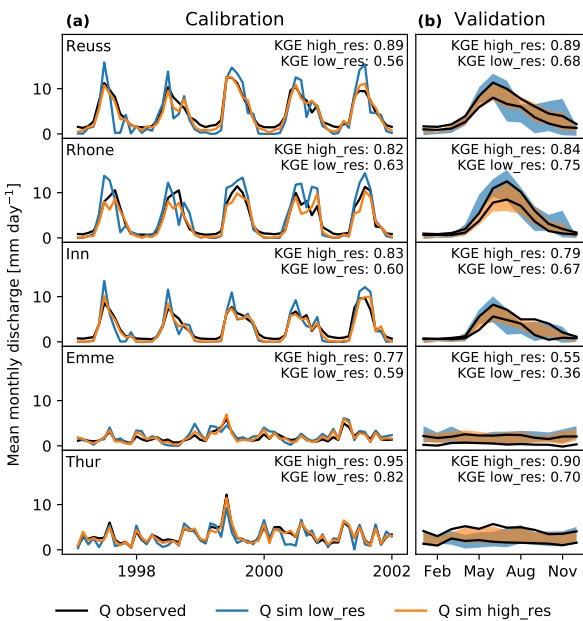

**Figure 5.** Discharge observations compared with discharge simulations for (a) the calibration period and (b) validation based on the monthly average discharge. In the top right corner of each subplot in (a) the Kling-Gupta efficiency (KGE) is presented. The range in (b) is plotted as the standard deviation around the mean monthly discharge, where the black lines indicate the lower and upper (mean +/- standard deviation) observed monthly discharge. Kling-Gupta efficiencies in these subplots are calculated over the entire simulation period, excluding the calibration years.

the mean matches better than the low resolution model. Overall, the low resolution model is able to accurately simulate these basins, yet the lack of spatial variability ensures that the high resolution model is able to reach better performance.

Hydrological response maps for the two main hydrological fluxes (actual evapotranspiration (ET) and generated runoff) during each extreme season are presented in Fig. 6. Grid cells are colored by their cell-specific standardized anomalies. ET anomaly maps are only shown for spring and summer periods, when this flux is most important. During the two other seasons, large parts of the basins are covered with snow, where the model assumes no ET to occur. The same maps on a monthly time step can be found in the Supplement. To validate how well these values represent the actual hydrological response, we compared the output from the high resolution model with observations from the research catchment Rietholzbach, situated within the Thur basin (see the black dot in Fig. 6). Evaporation observation was obtained from a long-term research lysimeter, and runoff was obtained from discharge observations from this catchment (Seneviratne et al., 2012). Both discharge and evaporation from the corresponding pixel were extracted from SPHY, to compare with the observations. We calculated the anomalies over the entire simulation period. The comparison between the observed and simulated anomalies can be found in Table 2. This table shows that the simulated anomalies agree well with the direction and magnitude of the observed anomalies. Winter and autumn values

**Table 2.** Comparison between anomalies simulated with SPHY and observed anomalies in the Rietholzbach, anomalies are based on the entire simulation period.

| Event | Runoff | | Evaporation | |
|-------|--------|--------|--------|--------|
| | Observed | Simulated | Observed | Simulated |
| DJF 1995 | 1.68 | 2.34 | 0.86 | 0.53 |
| MAM 2007 | -0.52 | -0.27 | 1.61 | 0.98 |
| JJA 2003 | -2.15 | -2.17 | 1.66 | 3.61 |
| SON 2002 | 2.65 | 2.62 | -1.76 | -0.26 |

for evaporation are gray, since they are not the focus of this study due to the fact that SPHY does not allow for evaporation during snow covered periods. There is a slight mismatch between the evaporation anomalies during the summer of 2003, yet both describe unusually high values. This mismatch can be attributed to the scale difference between the lysimeter and a single high resolution SPHY pixel, and the fact that SPHY does not account for all factors influencing evaporation since it uses the temperature-based Hargreaves method.

In Fig. 6, all basins show roughly the same ET response to the warm spring conditions in 2007. In the areas with a standardized anomaly of exactly zero, no evapotranspiration was simulated since the cells were covered with snow. Cells close to this region show a particularly high standardized anomaly. These cells are free of snow only for a limited time during spring, distorting the mean and standard deviation used to calculate the standardized anomaly. A more complex response is visible during the extremely warm and dry summer of 2003. In three basins, cells at low elevations show a different anomaly sign than the cells at mid/high elevations. In the entire region, higher temperatures increased the potential evapotranspiration, yet cells with a negative anomaly evaporated less water than normal. This indicates that those cells became water-limited during the course of the summer, and could no longer meet the potential ET. Cells at high elevations were able to meet the increased potential ET, and evaporated a lot more water than normal. This lead to a situation in which both negative and positive anomalies are present within the same basin, even at seasonal timescale and in response to a rather homogeneous distribution of temperature anomalies. Only the Rhone and Inn basins did not show this behavior, indicating that the low elevation cells did not become water-limited over the course of this summer.

Anomalies in the generated runoff also show a contrasting within-basin response, in particular in the Alpine basins. Here, cells with low elevations show a different anomaly than the cells at high elevations. This dependency between anomaly and elevation is not visible in the pre-Alpine basins, where all model cells show roughly the same response. The cause of this difference between the two basin types will be further investigated below in Fig. 8. Previously, we mentioned that the unusually wet autumn of 2002 was mainly due to a period with unusually high precipitation in November. The anomalies of the other seasons were mainly cause by a succession of multiple months with unusual temperature and/or precipitation values, so we chose to use a consistent time scale of three months throughout the paper. We also analyzed the hydrological response on a monthly timescale, but concluded that the response maps for November 2002 were not too different from the response maps for the autumn of 2002 (see Supplement).

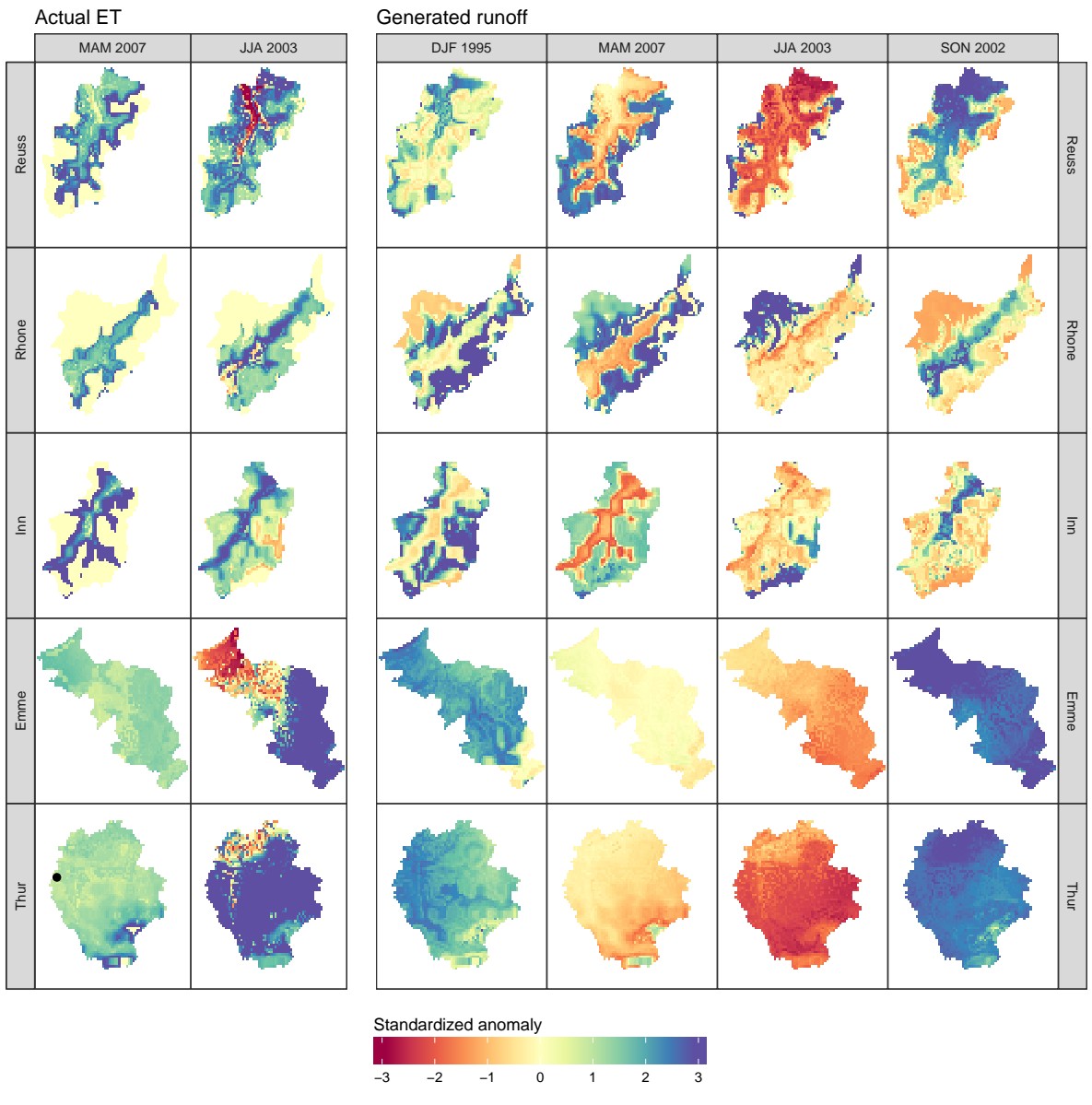

**Figure 6.** Spatial distribution of anomalies of actual evapotranspiration (a) and generated runoff (b) during the four extreme seasons, for all basins. The location of each catchment can be found in Fig. 1. Each box represents a size of ~40×40 km. The black dot in the Thur basin represents the location of the Rietholzbach research catchment.

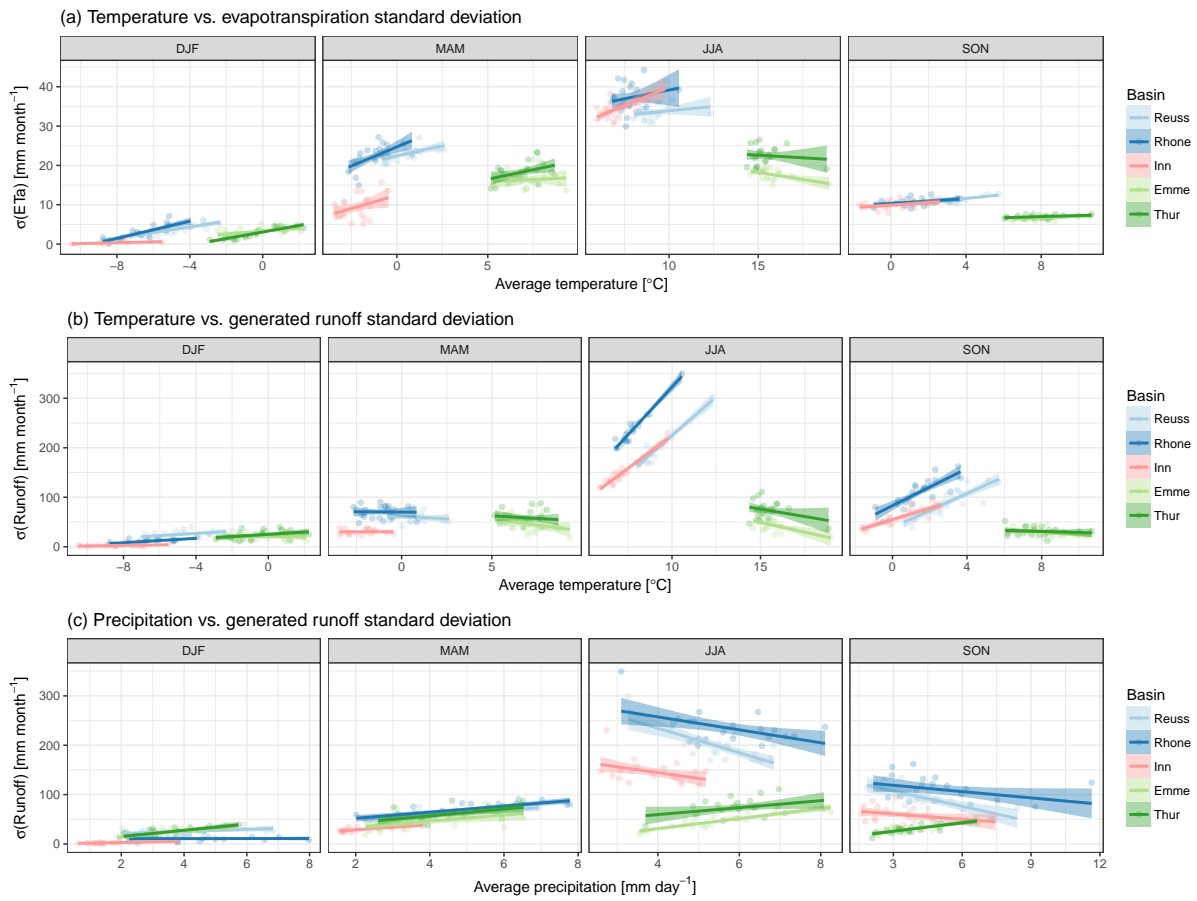

**Figure 7.** Relation between spatial standard deviation ($\sigma$) of simulated hydrological response and basin-averaged weather conditions: temperature versus evapotranspiration $\sigma$ (a), temperature versus runoff $\sigma$ (b), precipitation versus runoff $\sigma$ (c). Each point represents a single season in the 1994-2014 period. A linear regression through these points is represented as a solid line, with the shaded area indicating the 95% uncertainty range.

The spatial variability (as expressed by the standard deviation, $\sigma$) of both fluxes is plotted against the average forcing for all seasons in Fig. 7. Here the standard deviation is used as a measure of complexity, with large $\sigma$ values indicating a highly spatially variable and thus complex hydrological response. This figure gives insight into how the response complexity varies with basin average forcing. The precipitation - evapotranspiration plot was excluded since the graph consisted of random scatter, without a clear relation.

Spread in the actual evapotranspiration response seems related to temperature (Fig. 7a), where higher temperatures result in larger ET standard deviations. As mentioned earlier, potential evapotranspiration will increase with higher temperatures, but so does the number of water stressed cells. This combination increases the spatial $\sigma$ for evapotranspiration, and is visible in almost all basins and seasons.

Standard deviation of generated runoff seems most sensitive to temperature during summer and autumn, see Fig. 7b. The two catchment types show a different response: the runoff $\sigma$ increases with temperature in the Alpine basins, while runoff $\sigma$ decreases with temperature in the pre-Alpine basins. The cause for this difference is the presence of glaciers: glacier melt will increase with higher temperatures, while regions without glaciers will evaporate more. This contrast results in an increasing $\sigma$ with temperature in the Alpine basins, and in a decreasing $\sigma$ with temperature in the pre-Alpine basins. Please note that the average temperatures in both catchment types show hardly any overlap, making it difficult to identify how the basins would respond to the same temperature values.

Influence of average precipitation on the runoff $\sigma$ seems smaller (Fig. 7c). However, in the simulation period we selected, there is a correlation between temperature and precipitation. During winter, only the pre-Alpine basins show a response in runoff $\sigma$ to precipitation. The lack of response in the Alpine basins is related to temperature: the average winter temperatures in these basins hardly reaches values above $0°C$, where precipitation will fall as snow and does not directly contribute to runoff. A more pronounced relation between precipitation and runoff $\sigma$ is visible in summer and autumn, where $\sigma$ in the Alpine basins decreases with increasing precipitation and vice versa in the pre-Alpine basins. However, the Alpine regression lines are strongly influenced by the extremely warm and dry summer of 2003: without this season, the regression lines would have been much more horizontal. Since there is only one season this extreme in the 21 years of simulations, it remains difficult to separate the effects induced by temperature or by precipitation. The autumn period shows a similar response as the summer months, but the relation with temperature needs to be taken into account again. As visible in Fig. 4, seasons with unusually high precipitation are often related to lower temperatures, while seasons with less precipitation are often paired with higher temperatures; independent of the basin. This could indicate that the relation between precipitation and runoff $\sigma$ might be the inverse of the temperature - runoff $\sigma$ relation.

To gain a better understanding about the hydrological behavior within each basin, the standardized anomalies of each individual grid cell are plotted against elevation in Fig. 8. We again only show results for one basin of each catchment type, since the response patterns were similar across the different basins. The forcing anomalies show very little spread: the 95% confidence interval is almost always thinner than the plotted line, making it barely visible. Spread in runoff anomalies is bigger than the spread in forcing anomalies in both catchment types, making it impossible to explain the hydrological response solely by the forcing anomalies. Each dot in Fig. 8 is colored by land cover. Land cover shows a clear correlation with elevation, best visible in the Alpine basin. The pre-Alpine basins did not contain any glacier cells and only a limited number of sparse/bare cells. This is explained by their more limited elevation range compared to the Alpine basins (see Fig. 1b).

The hydrological response can be grouped per land cover class: "forest" and "glaciers" show nearly always a different response within the same basin and season, where "grass" and "other" are covering a gradual transition between the two groups. This grouping can be explained by the runoff generating processes. Areas at high elevation generate runoff by melting ice and snow (if present), while areas at low altitudes rely on rootzone and/or groundwater processes. The latter are mostly driven by the amount of available water (water limited), while runoff from ice and snow is mostly dependent on the incoming energy (energy limited). This dependency is best visible in Fig. 8a, where the hydrological anomalies at lower elevations coincide with the sign and size of the precipitation anomaly, while hydrological response shifts towards the temperature anomaly at

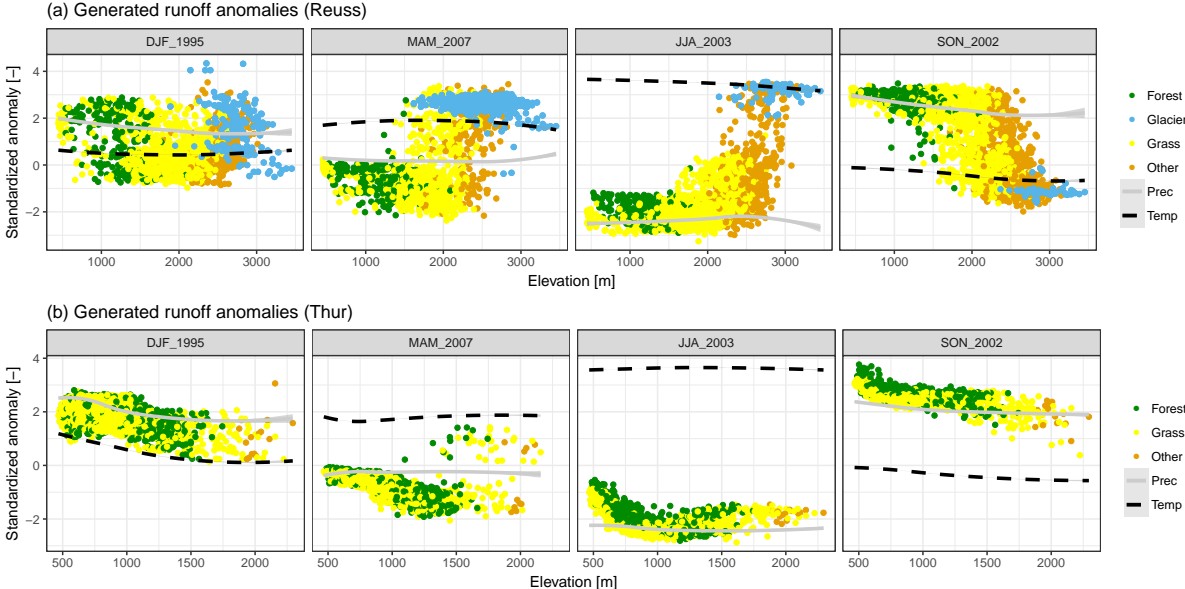

**Figure 8.** Relation between elevation and hydrological response colored by land cover type, presented for the Reuss (a) and Thur (b) basins. Each point represents the standardized anomaly for a single model cell, based on the data in Fig. 6. The solid and dotted lines show the smoothed precipitation and temperature anomalies, with the shaded area showing the 5−95% data range. Land cover type "other" represents all sparse and bare vegetation types.

higher elevations. Due to the insufficient "other" and "glacier" cells in the pre-Alpine basin, this relation is not as evident as in the Alpine basin. In the pre-Alpine basin, runoff anomalies seem to follow precipitation anomalies, indicating that the runoff generating processes are mostly driven by available water (Fig. 8b). This grouping of different responses matches with different zones defined by Theurillat and Guisan (2001): colline, < 700 m; montane, 700−1400 m; subalpine, 1400−2100 m;

5 alpine, 2100−2800 m; nival, > 2800 m. These zones match with the different land cover classes defined in our study: the first class is not represented in basin Reuss, montane corresponds to the "forest" group, subalpine to the "grass" group, alpine and nival to the "other" and "glacier" groups. A study by Jolly et al. (2005) described that these zones could also be used to group vegetation responses to the extreme summer of 2003. Furthermore, Fatichi et al. (2015) showed that changes in discharge as result of climate change show a clear relation with elevation, where catchments with high average elevation are expected to see

10 the biggest decrease in mean discharge, while catchments with low average elevation are expected to see a small increase in mean discharge. Our results combined with these studies indicate that elevation and thus vegetation cover are controlling the hydrological response to extreme seasons.

    Our results may be influenced by parameterizations defined within the model. For example, the limited evapotranspiration of snow-covered cells is a choice made by the developer of SPHY. One could argue whether this is realistic. Furthermore,

15 the glaciers in SPHY are fixed in location and extent. The importance of dynamical glaciers is investigated by Van Tiel et al. (2018), and they conclude that using a dynamical glacier module is most important for long term studies. The simulation period

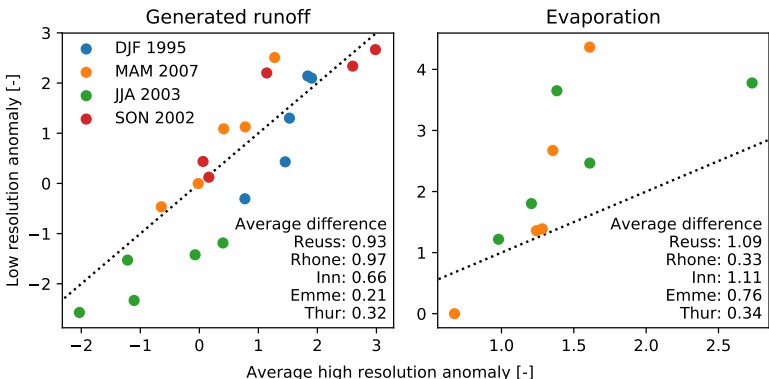

**Figure 9.** Comparison between the average high resolution model response and the low resolution model response, for the generated runoff. Colors indicate the different extreme seasons, and the dotted line represents the 1:1 line.

of our study was rather short, and we therefore expect only minor differences in the location and extent of the glaciers over our time period. We do not expect any major different results and conclusions as result of those parameterizations within SPHY.

## 3.2 Impact of model resolution

With improved understanding of the hydrological response to extreme seasons when simulated at high resolution (matching
the regional scale studies), we can now compare those results to the model output when the basins are simulated on a $0.5° \times 0.5°$ resolution (matching the global scale studies). Firstly, we compare how well the aggregated high resolution response corresponds with the low resolution model, see Fig 9. All pixels within the high resolution model are averaged and compared with the anomaly calculated for the low resolution model. Ideally, the low resolution model should match the aggregated high resolution model response. This figure shows that generally both models simulate the same trend, yet the order of magnitude
of the anomaly does not always match. The presented average difference represents the mean absolute difference between the high and low resolution model results. This value shows that the resolution difference generally causes a bigger disagreement in the Alpine basins than in the pre-Alpine basins. Overall, the runoff simulated with the low resolution model matches the high resolution model relatively well. This is in line with the conclusions from Kling and Gupta (2009), who stated that lumped models are able to reach similar runoff predictions as a distributed model. However, when investigating local responses, the
prediction from the low resolution model might not be representative.

Next, we compare how the range of values from the high resolution model compare to the low resolution model in Fig. 10. In this figure, output from only two basins is shown since results were similar across basins of the same catchment type. High resolution model responses are clearly not normally distributed, but have a bimodal or skewed distributions. Response of the pre-Alpine basin shows less variation than the Alpine basin, which was also visible in Fig. 8. In all cases, the low resolution
model anomaly is within the high resolution model anomaly range, but does not show a consistent position within this range. This figure makes it difficult to quantify the differences between the low and high resolution models.

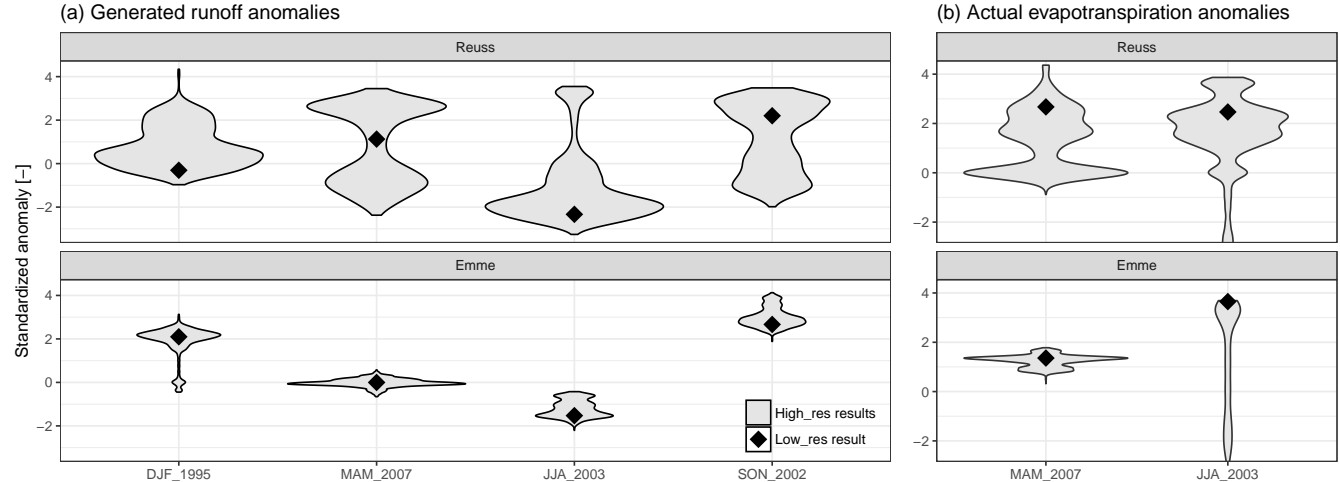

**Figure 10.** Model response to extreme seasons for both generated runoff (a) and actual evapotranspiration (b), where violin plots represent the high resolution model response and the diamond the low resolution model response.

**Table 3.** Scale mismatch between the high and low resolution models as measured by DWD, for both hydrological fluxes during the four extreme seasons.

| Basin | **Total generated runoff** | | | | **Actual ET** | |
|---|---|---|---|---|---|---|
| | DJF 1995 | MAM 2007 | JJA 2003 | SON 2002 | MAM 2007 | JJA 2003 |
| Reuss | 2.63 | 2.31 | 4.81 | 2.58 | 2.44 | 3.70 |
| Rhone | 2.86 | 2.82 | 4.29 | 1.88 | 0.85 | 1.52 |
| Inn | 2.40 | 1.82 | 4.56 | 1.84 | 4.36 | 2.19 |
| Emme | 1.25 | 0.33 | 0.79 | 1.01 | 0.46 | 5.96 |
| Thur | 0.97 | 0.73 | 1.48 | 0.66 | 0.62 | 4.33 |

For each hydrological flux, basin and extreme season, the Density Weighted Distance (DWD) is calculated and presented in Table 3. This table shows that the runoff DWD in the Alpine basins is generally higher than the DWD in the pre-Alpine basins (average Alpine $DWD = 2.90$ and average pre-Alpine $DWD = 0.90$). This is also visible in Fig. 10, where the pre-Alpine runoff violin plots cover a smaller anomaly range than the Alpine violin plots. These averages indicate that the high 5 resolution model anomalies can deviate with 2.61 and 0.81 standardized anomalies from the low resolution model anomaly in the Alpine and pre-Alpine basins, respectively. This illustrates that in these areas, the local hydrological response can be a lot more extreme than the low resolution model might indicate. This effect is largest in the Alpine basins, which can be explained by the wider range in elevation and land cover types.

The summer of 2003 in the Rhone basins shows a very high DWD value for the generated runoff. This is due to a combination 10 of a relatively low percentile score ($P = 0.18$) and a large distance to the upper 95% anomaly ($d_{upper} = 5.57$). A very large

portion of the high resolution model values is close to the low resolution model anomaly, implying that a small increase in low resolution model anomaly would significantly increase the $P_{\text{low\_res}}$ values, which would have reduced the emphasis on $d_{\text{upper}}$, decreasing the DWD value, see Fig 10b.

Another high DWD value is found for actual evapotranspiration during the summer of 2003 in the Emme basin (see Fig. 10b). The high resolution model results show a long tail towards negative anomalies, caused by model cells which are water-limited during this season. The low resolution model is not able to replicate the response, since the model consisted of only a single grid cell. This cell was not water-limited during this season, since higher than average ET was simulated. As a result, the low resolution model is not able to mimic basin responses which are as far as 5.66 standardized anomalies away from the low resolution model.

Actual evapotranspiration is not only dependent on the amount of available water, but snow cover is also an important factor. For example. the high DWD value for evaporation in the Inn basin during the spring of 2007 can be attributed to this response. In the low resolution model, the cell was free of snow, allowing the model to evaporate, while in the high resolution model only half the cells were free of snow. The cells covered with snow were not able to evaporate water, resulting in a large variation in anomalies and thus a large $5-95\%$ range.

Our results are in line with numerous studies either investigating effects of model resolution or comparing the performance of lumped models with (semi-)distributed models. For example, Leung and Qian (2003) studied the sensitivity of simulation results to model resolution, and concluded that the high resolution model was able to better represent the spatial variation than the low resolution model. Gao et al. (2006) concluded that the simulations improved as model resolution increased, since the local dynamics are better represented in the model. However, as stated by Lucas-Picher et al. (2012) and Pryor et al. (2012), it is not given that high resolution simulations always lead to better results, as it becomes challenging to validate the model results with observations, especially at fine spatial resolutions and/or with large spatial coverage. However, they state that the model might become more physically plausible if complex processes are better represented at these scales. As showed by Lobligeois et al. (2014), correct representation of the spatial patterns in precipitation can strongly influence the quality of the simulations in basins with a lot of spatial variation in precipitation. Boyle et al. (2001) concluded that improvements in model performance were related to the spatial distribution of the model input. Koren et al. (2004) reached a similar conclusion, stating that their distributed model outperformed the lumped model in basins with significant spatial rainfall variability. Finally, Carpenter and Georgakakos (2006) compared a lumped model with a distributed model and concluded that the gain in performance was dependent on the amount of spatial variation present in the region of interest. Our study showed that the difference between the high and low resolution simulations is largest in basins with large spatial variability. In our study, we show that also the dominant runoff generating processes are an important factor for the differences between the low and high resolution model.

The results may be influenced by the fact that the model did not allow for sub-grid variability in land use or soil types, something other models might have included. When sub-grid variability is taken into account, we expect the low resolution model results to become less extreme. However, the low resolution model will not be able to capture the full dynamics simulated with the high resolution model, since landscape characteristics still need to be aggregated to a coarser resolution.

## 4 Summary and conclusions

In this study, we investigated the hydrological response anomalies in five catchments in the Swiss Alps at two different spatial resolutions. The catchments were selected based on topography and land cover. Three out of five catchments are situated at high elevations and contain glaciers (referred to as Alpine catchments), and the two other catchments are situated at lower elevations and do not contain glaciers (referred to as pre-Alpine basins). We ran the distributed hydrological model Spatial Processes in Hydrology (SPHY) at two different spatial resolutions to match two common hydrological modeling approaches: at a high resolution of ~500×500 m to match regional scale studies (and matching "hyperresolution"), and at a lower resolution of ~40×40 km to match global scale studies performed at 0.5×0.5° resolution. Model results were aggregated per season, and were analyzed based on standardized anomalies. For each season, we selected one season with unusual precipitation and/or temperature values within the simulation period of 1993-2014: winter of 1995, spring of 2007, summer of 2003 and autumn of 2002.

Results from the high resolution model show that the intra-basin response covers a large range of anomalies during the selected seasons, where contrasting anomaly signs within a single catchment are often occurring. Within-basin complexity of hydrological response was found to generally increase with the magnitude of the forcing anomaly. The low resolution model failed to capture this diverse and contrasting response, since the entire region was covered by a single grid cell. The newly introduced Density Weighted Distance (DWD) was used to quantify the variability simulated with the high resolution model that is missed by the low resolution model. The DWD indicated that the local response differed on average more than 2 standardized anomalies from the response simulated with the low resolution model. Our results show that results generated with a high resolution model are not only more variable, but anomalies can locally be much more extreme or even of the opposite sign than a low resolution model might indicate. This conclusion confirms previous results by Melsen et al. (2016a), who found that results of large-domain models should be interpreted with care because of a lack of spatial variability in these models. Since our low resolution model did not represented sufficient spatial variability, this led to a large discrepancy between the high and low resolution model results.

The variability in simulated response was associated with the different land cover classes. We found that runoff anomalies were matching the temperature anomalies when the dominant runoff generating processes are energy-limited (snow/glaciers), and runoff anomalies were matching precipitation anomalies when the dominant runoff generating processes are water-limited (grass/forest). The two pre-Alpine basins generally showed a different response than the Alpine basins, which can be attributed to the smaller variation in elevation and land cover in these basins. The grouping of responses in our study matches the elevation classes as defined by Theurillat and Guisan (2001).

*Code and data availability.* SPHY model code (version 2.1) is available at https://github.com/FutureWater/SPHY/tree/SPHY2.1, Digital Elevation Model is available at http://srtm.csi.cgiar.org, discharge data was obtained from https://www.hydrodaten.admin.ch/en/stations-and-data. html, land cover data was obtained from https://land.copernicus.eu/pan-european/corine-land-cover, soil data was obtained from https:

//daac.ornl.gov/SOILS/guides/HWSD.html, distributed forcing data (precipitation and temperature) are archived by the Swiss Federal Office for Meteorology and Climatology (MeteoSwiss).

*Author contributions.* JB, AJT, and RH designed the research. JB performed the research, analyzed the data, and wrote the draft; all authors contributed to interpreting results, discussing findings, and improving the manuscript.

5   *Competing interests.* The authors declare that they have no conflict of interest.

*Acknowledgements.* The authors would like to thank the editor Nadav Peleg, and Davide Zoccatelli, Staffan Druid and the anonymous reviewer for their constructive comments, which helped to improve the quality of this paper.

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
