# Peer review of "Evaluating seasonal hydrological extremes in mesoscale (pre-)Alpine basins at coarse $0.5^{\circ}$ and fine hyperresolution"

_Hydrology and Earth System Sciences, 2018_

## Referee Comment (RC1) · D. Zoccatelli (Referee) · 29 Oct 2018

The authors present a comparison between the application of a model at two different scales in terms of discharge and evapotranspiration response to seasonal extremes. The main conclusions are that the coarse resolution model fails to represent the complex internal response of Alpine basins, and that hydrological response can locally be significantly more intense than what predicted at coarse resolution. The paper presents an interesting application, but with few major flaws in the presentation and in the analysis. For this reason, I suggest that the paper could be accepted after major revision.

Main comments:

[Figure]

- One of my main concerns is the metric introduced: DWD. High values of this statistic may mean that the low_res model performances are far from the high_res OR that the high_res model has high internal variability. Even if low_res is perfectly representing the cumulated catchment response, if the high_res has high variability you will still have high DWD results. This has strong implications on your results. P17L21: "the low resolution model misses on average more than 2 standardized anomalies compared to the high resolution model." To me sentences like this are confusing: the distributed model range is about 2 standardized anomalies away, but that's not the low_res model fault in any way. Another concern about the DWD metric: if we consider DWD, as you say "a difference in terms of number of standardized anomalies" then we can consider P_low-0.05 and 0.95-P_low as weights. The sum of these weights however is different from 1 and it is a function of the threshold chosen. The consequence, as you point out, is that this statistic is very affected by the threshold chosen. By absurd, if we take the range around the 50th percentile DWD would always be = 0.

- If the point above is valid, right now the main conclusion of the paper is that the low_res model fails to describe the internal variability of the catchment, which seems trivial. When I first read the paper I was expecting more a comparison between the catchment-scale response from the two models, and if the low_res model is able to capture the aggregated catchment response during seasonal extremes. Because of the metric used, this is shadowed by the failed representation of the internal variability. In addition to the current analysis, I think that the results aggregated at catchment scale should also be compared.

- As you say, scale issue in hydrological models is a major issue, and has been extensively studied. However, you give completely no background about this field of research which, I think, is very closely related with your objectives and results. I think that this aspect should be emphasized both in the introduction and in the discussion.

Hydrological model issues:
- There is no verification of the intra basin response, only on the outlet discharge. Still, you take the internal variability of discharge and evapotranspiration as reference. Some verification would significantly strengthen the results.

- Related to the point above: in your model is each cell independent? In steep catchments soil moisture in the valley is often fed by the hillslopes. A result like fig.6, where higher cells evaporate more while lower cells are water-limited and evaporate less, can be a direct consequence of this model limitation. This could have important consequences on your current results.

- In the model description you don't talk about runoff propagation, but in calibration you compare runoff measured at the outlet. Does your model include runoff propagation?

- I don't understand why you calibrate on the square of residuals and then use KGE to evaluate model simulations. Why you don't use KGE from the begin?

- No statistic on validation is presented, despite having 21 years of data. Why?

Minor comments:

- P12L26: to me more ET does not necessarily result in more uniform response across the basin.

- Fig 7: dangerous to compare a dimensional variable between basins with different mean annual precipitation. Also, table 1 is missing info on mean annual precipitation.

- Figure 4: where is the simulated anomaly in hydrological response mentioned in the caption?

---

## Short Comment (SC1) · 30 Oct 2018

Dear Davide Zoccatelli,

Thank you very much for your constructive review. The comments made are valid and we agree that we need to clarify some parts of the manuscript to improve the paper. We will discuss how to address your comments (by updating and/or adding figures/tables), and in the coming days, we will propose changes to our manuscript in order to resolve the comments made in your review. After we post this, we really appreciate it if you could assess whether the proposed changes clarify the points from your review.

[Figure]

In the meantime, thanks again for your contribution to our manuscript.

With best regards, Joost Buitink, on behalf of all authors

---

## Author Comment (AC1) · 7 Nov 2018

**Detailed response to Davide Zoccatelli's comment**

Joost Buitink, Remko Uijlenhoet, and Adriaan J. Teuling

**1 Density Weighted Distance**

One the main comments by Davide Zoccatelli was related to the use of the new metric that was introduced in the manuscript, the Density Weighted Distance (DWD). Davide Zoccatelli made several valid comments, which we will address below.

- Davide Zoccatelli correctly mentioned that high DWD values can mean two things: either that the low_res model result is outside of the range of values simulated with the high_res model, or that the high_res model has high internal variability. This is indeed a property of this method, which we will explicitly mention when we introduce this method. If the low_res model results is different than the entire range of values from the high_res model, we recommend not to use DWD since this is not the intended situation to apply DWD. To prevent this, all conclusions related to DWD values are not solely based on the DWD results, but are interpreted in combination with the violin plots, which proved the necessary additional information. We agree that this is currently not clearly presented as such in the manuscript, and we will make sure this is clarified in the new version of the manuscript.

- DWD can give high values, even if the low_res model is perfectly representing the accumulated high_res results. This intended behavior of the method, since we want to quantify how well the low_res model represents the entire high_res model results. However, we do agree that a comparison between the low_res model and the aggregated catchment response from the high_res model is currently missing from the manuscript. In Figure 1 we present a comparison between the aggregated high resolution model response and the response from the low resolution model. All pixels within the high resolution model are averaged and compared with the anomaly calculated for the low resolution model. Ideally, the low resolution model should match the aggregated high resolution model response. This figure shows that generally both models simulate the same trend, yet the order of magnitude of the anomaly does not always match. The presented average difference represents the mean absolute difference between the high and low resolution model results. This value shows that the resolution difference generally causes a bigger disagreement in the Alpine basins than in the pre-Alpine basins.

- Davide Zoccatelli correctly noted the the $P_{\mathrm{low\_res}} - 0.05$ and $0.95 - P_{\mathrm{low\_res}}$ can be interpreted as weights, and that the sum of those is different from 1. Therefore, we propose a new slight modification of the DWD, based on the following

[Figure]

**Figure 1.** Comparison between the average high resolution model response and the low resolution model response, for the generated runoff. Colors indicate the different extreme seasons, and the dotted line represents the 1:1 line.

equations:

$$DWD = W_{\text{lower}} \cdot d_{\text{lower}} + W_{\text{lower}} \cdot d_{\text{lower}}, \tag{1}$$

$$W_{\text{lower}} = \max\left(0, \min\left(1, \frac{P_{\text{low\_res}} - P_{\text{lower}}}{P_{\text{upper}} - P_{\text{lower}}}\right)\right), \tag{2}$$

$$W_{\text{upper}} = \max\left(0, \min\left(1, \frac{P_{\text{upper}} - P_{\text{low\_res}}}{P_{\text{upper}} - P_{\text{lower}}}\right)\right), \tag{3}$$

$$d_{\text{lower}} = Z_{\text{low\_res}} - Z_{\text{high\_res}}^{5\%}, \tag{4}$$

$$d_{\text{upper}} = Z_{\text{high\_res}}^{95\%} - Z_{\text{low\_res}}, \tag{5}$$

where the $W_{\text{lower}}$ and $W_{\text{upper}}$ replace the original $P_{\text{low}} - 0.05$ and $0.95 - P_{\text{low}}$ terms, respectively. Both weights are corrected between 0 and 1, which is only necessary when $P_{\text{low\_res}}$ is outside the $P_{\text{lower}} - P_{\text{upper}}$ range. This has some minor implications for the DWD results, since the distances are now weighted with a total weight of 1, instead of 0.9 in the previous version. This has the effect that the DWD values are slightly higher. The absolute DWD values will change, but it has no implications for the overall conclusions, see Figure 2 and Table 1. These results are a function of the chosen threshold, yet we would recommend to choose $P_{\text{upper}}$ and $P_{\text{lower}}$ to include most of the high resolution data, since every pixel can be considered equally important. We are happy to learn if this modified definition solves the original concerns raised.

**2 Hydrological model validation**

– Davide Zoccatelli also questioned the validation of the model. Originally, our validation was only shown in Figure 5b (original manuscript). We think this might have easily been overlooked. Therefore, we have updated our calibration/validation figure to better represent the validation of the model, and added KGE values as model validation. In

[Figure]

**Figure 2.** DWD examples with the updated DWD equations.

**Table 1.** Scale mismatch between the high and low resolution models as measured by DWD, for both hydrological fluxes during the four extreme seasons (based on the updated DWD equations).

| Basin | Total generated runoff | | | | Actual ET | |
|---|---|---|---|---|---|---|
| | DJF 1995 | MAM 2007 | JJA 2003 | SON 2002 | MAM 2007 | JJA 2003 |
| Reuss | 2.63 | 2.31 | 4.81 | 2.58 | 2.44 | 3.70 |
| Rhone | 2.86 | 2.82 | 4.29 | 1.88 | 0.85 | 1.52 |
| Inn | 2.40 | 1.82 | 4.56 | 1.84 | 4.36 | 2.19 |
| Emme | 1.25 | 0.33 | 0.79 | 1.01 | 0.46 | 5.96 |
| Thur | 0.97 | 0.73 | 1.48 | 0.66 | 0.62 | 4.33 |

original figure, calibration and validation data for the low resolution model was missing, so we added this to the new version of this graph, see Figure 3. We hope this figure gives a better overview of the model performance.

– Davide Zoccatelli also noted that there is no internal validation of the model. To get an idea of the quality of the internal flux representation, we compared the simulated fluxes with observed data from the research catchment Rietholzbach, situated within the Thur basin (Seneviratne et al., 2012). We obtained the evaporation data as measured with a long-term research lysimeter, and discharge data from this catchment. Both discharge and evaporation from the corresponding pixel were extracted from SPHY, to compare with the observations. We calculated the anomalies over the entire simulation period. The comparison between the observed and simulated anomalies can be found in Table 2. This table shows that the simulated anomalies agree well with the direction and magnitude of the observed anomalies. Winter and autumn

[Figure]

**Figure 3.** Calibration and validation of SHPY. The model is validated over the entire period, excluding the calibration period. The two black lines, and the shaded areas present the average discharge plus/minus the standard deviation. For Inn, all data after 2003 was excluded, since the observed discharge pattern changed after this period.

values for evaporation are gray, since they are not the focus of this study due to the fact that SPHY does not allow for evaporation during snow covered periods. We believe that this table is a valuable addition to the validation of the model.

**Table 2.** Comparison between anomalies simulated with SPHY and observed in the Rietholzbach, anomalies are based of the entire simulation period.

| Event | Runoff | | Evaporation | |
|---|---|---|---|---|
| | Observed | Simulated | Observed | Simulated |
| DJF 1995 | 1.68 | 2.34 | 0.86 | 0.53 |
| MAM 2007 | -0.52 | -0.27 | 1.61 | 0.98 |
| JJA 2003 | -2.15 | -2.17 | 1.66 | 3.61 |
| SON 2002 | 2.65 | 2.62 | -1.76 | -0.26 |

There is a slight mismatch between the evaporation anomalies during the summer of 2003, yet both describe unusually high values. To further investigate this, we also plotted the evaporation time series in Figure 4. This figure shows that

SPHY is able to accurately simulate the evaporation, yet there are some differences between the two time series. These can be attributed to the scale difference between the lysimeter and a single high resolution SPHY pixel, and the fact that SPHY does not account for all factors influencing evaporation since it uses the temperature-based Hargreaves method. The difference between the two time series influences the mean and standard deviation, and therefore the resulting anomaly values.

[Figure]

**Figure 4.** Comparison between observed evaporation in a lysimeter, and the simulated evaporation of the corresponding pixel.

**3   Comments and other changes**

– We will add average yearly precipitation to Table 1 in the original manuscript.

– We will add more references and background information to the manuscript regarding scaling issues in hydrology/hydrological modeling.

10   – SPHY assumes that each model pixel is independent, meaning that there is no communication between the individual pixels. This is a limitation of the model, however most (conceptual) hydrological models are programmed this way. We believe the impact on the results to be small.

– SPHY does include runoff propagation, where runoff is transported to the downstream cells using a recession coefficient. For the exact details for the routing conceptualization, we refer to the paper of Terink et al. (2015).

15   – We calibrated the model based on the sum of squares, since this is the more common approach to model optimization However, KGE is easier to interpret than the square of residuals.

**References**

Seneviratne, S. I., Lehner, I., Gurtz, J., Teuling, A. J., Lang, H., Moser, U., Grebner, D., Menzel, L., Schroff, K., Vitvar, T., and Zappa, M. (2012). Swiss prealpine Rietholzbach research catchment and lysimeter: 32 year time series and 2003 drought event. *Water Resources Research*, 48(6):W06526.

Terink, W., Lutz, A. F., Simons, G. W. H., Immerzeel, W. W., and Droogers, P. (2015). SPHY v2.0: Spatial Processes in HYdrology. *Geosci. Model Dev.*, 8(7):2009–2034.

---

## Short Comment (SC2) · 10 Nov 2018

**PEER REVIEW**

**Introduction**

As part of the course Research trends in Physical Geography & Hydrology, students of the master's programme in Earth science at Uppsala University were given the task to perform a peer review of an article of their choice. This peer review was made by Staffan Druid, hydrology student at said program.

**Article**

The article chosen for this assignment is *Evaluating seasonal hydrological extremes in mesoscale (pre-)Alpine basins at coarse 0.5 and fine hyperresolution* by **Joost Buitink**, **Remko Uijlenhoet** and **Adriaan J. Teuling** of the Hydrology and Quantitative Water Management Group of Wageningen University in The Netherlands. The article was submitted to the journal Hydrology and Earth System Sciences (HESS).

The article can be found in its entirety at:
https://www.hydrol-earth-syst-sci-discuss.net/hess-2018-407/hess-2018-407.pdf

**Summary**

The use of models is a helpful, and in many cases efficient, means to understand and quantify processes in Earth sciences. Therefore it is important to be aware of their applications and limitations. The article delineates the effect of spatial resolution when simulating hydrological parameters (Runoff and Actual Evapotranspiration) in different Swiss river basins at two different resolutions (cells being $40 \times 40$ km or $500 \times 500$ m). Using a newly introduced metric of comparing the simulated anomalies of the hydrological parameters, the article finds that the high resolution models more accurately simulated the complexity of the anomalies in the basins, especially at seasons with extreme forcing in hydrological parameters. Furthermore, the low-resolution models failed to present the more extreme levels of anomalies compared to the high resolution models.

**Review**

Overall, the article is well-written and presents interesting results that are relevant for the understanding and choice of models for future research. After a few minor revisions of typos, slightly unclear paragraphs and further motivation of the importance of the results, this article could probably be accepted.

**Overall comments**

- The article is thoroughly written with a clear structure that is easy to follow. There is continuously good use of the equations, figures and examples that makes the reading interesting and concise. The article does not make assumptions on the reader's background knowledge, but rather explains too much than too little, which is a strong quality; making sure that the reader understands each step of the study is a way to increase the credibility and justifying your results.
- The choice of methods is well motivated and justified. For instance, the thorough explanation of the SPHY model (section 2.3) could have been deemed not as relevant for this article, but the given explanation helps the reader follow the reasoning behind the choice of that model, and ultimately interpreting the results. The same goes for the explanation of the DWD metric (section 2.5), where the prior presentation of alternative

metrics and discussion of their unsuitability strengthens the motivation of using the DWD. Again, one could argue that these alternative metrics are irrelevant for the study, but for the credibility of the results it makes sense to include them. Using figure 3 to visualize the concept of the DWD is important to understand the function the new metric.

- Although the introduction is clear, the abstract is somewhat confusing and overwhelming. Reading it did not give me a sense of the results or the relevance of the article, or introduced the topic or theme of the article. This makes the aim and method slightly diffuse and it's hard to grasp the full picture until reaching the end of the article. Condensing such an extensive article is never easy, but using less technical terms and slightly simpler language could improve understanding and motivate further reading. Of course, this depends on the target audience and desired impression of the authors.

- The results are well presented and seem reasonable, but are not surprising. When comparing models using thousands of cells to models comprised of one (1) single cell, there's bound to be less variation and detail in the latter. Although the low resolution is well motivated (as being common in global-scale models), the choice of basins that in that resolution only consist of one cell is less clearly motivated. Perhaps a somewhat higher resolution (giving the basins at least a few 10's of cells) would be more interesting and nuanced to compare to the high-resolution model results. While it's always important to validate assumptions that are taken for granted, it's hard to say to what extent the results of comparing such extremely different resolutions are useful in the sense of presenting new knowledge.

- References to other studies and up-to-date literature are present throughout the article. The structure is rational and clearly defined, while simultaneously well connected between sections. The language is concise, easy to follow and does not use overly complicated abbreviations or technical terms.

**Conclusion**

The article is clear, relevant and easy to follow. The use, and thorough explanation, of figures, equations and examples in combination with the excellent level of complexity of the language is the strongest quality of the article. A few minor typos and paragraphs with potential for clarification exist, but are easily fixed and may also just be a matter of personal taste. The abstract and overall aim of the article could be more clearly presented, as well as the motivation for selecting basins that generates models with only one cell. Additionally, the relevance of the (hardly surprising) results for future studies and for the scientific community as a whole could be more thoroughly presented.

**Trivial comments**

- Typos on page 8: line 5 reads "bimodel" instead of *bimodal*, line 10 reads "5-59% range" instead of (supposedly) 5-*95%* range.

- The four calibration parameters listed on page 6, lines 10-13 are a bit confusing. Since the third parameter ("a parameter determining the fraction of water that can refreeze in the snow pack"), unlike the others, is not given a name, one might confuse this as simply the description of the second parameter, thereby only listing three parameters in total. This could easily be made clearer by simply attributing the third parameter with a name and not only description.

- Specific basins in figures that are referenced in the text are sometimes not specified by name (e.g. page 9, line 5; page 11, line 1). By naming the basins explicitly in the text, it would be easier for the reader to see the relevant points in the figure as well.

- To further clarify the nature of the basins (Alpine or Pre-Alpine) to the reader, figures such as figure 6 could name the basins with a subscript of abbreviation of the type (such as "Reuss$_A$" for the Alpine Reuss basin or "Inn (PA)" for the Pre-Alpine Inn basin).

---

## Author Comment (AC2) · 12 Nov 2018

Dear Staffan Druid,

Thank you very much for your review and suggestions for improvement of our work. As we understood from your review, the main point is that the results do not appear to be surprising, as was also mentioned by the earlier review by Davide Zoccatelli. We can see that those results might not appear to be surprising at a first glance, since scale issues have been studied in hydrology for decades. However what is new in our study is the focus on quantifying how high and low resolution models capture climate extremes, thus bridging the climate perspective (where events are often expressed in

standardized anomalies) and the hydrological scale perspective including within-basin complexity. As far as we know, we are the first ones to show that is not uncommon to have both extreme positive and negative flux anomalies occurring simultaneously within a catchment. We do recognize that our aim might not be formulated clearly enough in the introduction and abstract, and we will ensure that this specific aim gets more focus in the next version of the manuscript.

Furthermore, thank you for your kind remarks and other suggestions for improvement, these will definitely help to further improve our manuscript.

With best regards,

Joost Buitink, on behalf of all authors

---

## Referee Comment (RC2) · Anonymous Referee #2 · 26 Nov 2018

The manuscript "Evaluating seasonal hydrological extremes in mesoscale pre-Alpine basins at coarse 0.5 deg and fine hyperresolution" by Buitink et al. provides an interesting excursion into the effects of spatial resolution in hydrological modelling. Two grid resolutions, 500x500 m and 40,000x40,000 m are used in the STAHY model to simulate hydrological processes, and the results are compared in 5 mesoscale basins in the Swiss Alps. The main message is that the coarse resolution model fails to capture the "diverse and contrasting response" from the high resolution model, because topography and land cover are not accurately represented. This is found to be especially true for extremes, where anomalies in climate and their effect on runoff and ET were quantified.

[Figure]

It has to be said that these conclusions are not suprising, in fact are to be expected, and there are numerous studies published in hydrological literature that report the same or similar findings. In this sense, the potential innovation of the paper is rather limited, and would have to be found in the details and/or implications that are specific to the study cathments, climates, model used, etc. To highlight what is really innovative and make the relevance of the paper more clear I suggest to focus in the revision on the following points and questions.

1. What is the real aim of the paper? In the introduction (p2, 10-15) the authors claim that many studies have explored the effect of spatial resolution, but few (none) have explored the "effect of the modelling approach". It should be clarified from the start what is meant by this, because the authors in my opinion also only show the effect of spatial resolution and not modelling approach. They use the same conceptual model, same parameterisations, only the input data differ.

2. The studied catchments are all smaller than one coarse pixel in the analysis. This is quite clearly stated in Section 2.4. The input data are resampled to the 500x500 grid and averaged to a 40,000x40,000 m grid for the coarse application. So this sounds to me like comparing a spatially distributed model to a point model, not to a coarse resolution model. This also means that all elevation dependencies in hydrological processes in the point application are gone. Is this correct?

3. I would have liked to see at least a table with the values of the key parameters that were calibrated, i.e. something that gives more credibility to the SPHY model application. It seems in Section 2.4 that the model was calibrated for both the spatial application and the point application separately. I assume it is the high resolution application shown in Figure 5. How different were the parameter values? What do the differences (if any) mean for the results of the simulations, e.g. temperature dependencies, etc.

4. The results were compared on seasonal anomalies of runoff and ET, summed over the catchment areas. There are no supporting data and plots to actually show how

the model performed for spatially distributed variables, beyond Figure 6. For example, snow cover could have easily been compared with data to show how snow accumulation and melt processes are simulated. There is little confidence given to the spatial predictions of the model,on which the entire analysis is based.

5. A new metric DWD is proposed to show the place of the point model value in the spatial distribution of the spatially resolved values. This is an interesting metric. I appreciate the effort to illustrate its use in Figure 3.

6. The relation of the seasonal anomalies in ET and runoff to temperature across the basins in Figure 7 is interesting. I have one concern that the results here are probably strongly dependent on the parameterisations and structure of the model. Some of these relations, e.g. between runoff and temperature can be gleaned directly from observations. Will you get the same sensitivities? In Figure 8 the anomalies are plotted for every grid cell as a function of land cover. What about soil depth? Does SPHY assume that soil depth and soil properties are constant in space?

7. Overall, I find the physical relations between the results and hydrological processes in Section 3.1 nicely covered, the arguments are logical, especially the elevation effect is coming out strongly. I suggest also looking at the paper by Fatichi et al. (2015) "High-resolution distributed analysis of climate and anthropogenic changes on the hydrology of an Alpine catchment" in Journal of Hydrology for another demonstraiton of this effect with a physically-based model.

8. As mentioned above, the resolution effects are less instructive than the explanation of the anomalies. I am not sure what to take out of Figure 9, other than the point model lies within the range of the cells of the distributed model. This was for an extreme year, what about the entire simulation? Probably the results are the same. The message simply seems to be that lumping in space averages hydrological response, which is something very well known. Is there more to it than that? If yes, this has to be brought to the forefront more clearly.

---

## Author Comment (AC3) · 28 Nov 2018

**Detailed response to Anonymous Referee #2**

Joost Buitink, Remko Uijlenhoet, and Adriaan J. Teuling

We would like to thank Anonymous Referee #2 for taking their time and effort to read our manuscript. The constructive comments will be used to improve our manuscript. Below, we will respond to the comments made by Anonymous Referee #2 (original comments in black, our response in blue).
* * *
The manuscript "Evaluating seasonal hydrological extremes in mesoscale pre-Alpine basins at coarse 0.5 deg and fine hyper-resolution" by Buitink et al. provides an interesting excursion into the effects of spatial resolution in hydrological modelling. Two grid resolutions, 500x500 m and 40,000x40,000 m are used in the STAHY model to simulate hydrological processes, and the results are compared in 5 mesoscale basins in the Swiss Alps. The main message is that the coarse resolution model fails to capture the "diverse and contrasting response" from the high resolution model, because topography and land cover are not accurately represented. This is found to be especially true for extremes, where anomalies in climate and their effect on runoff and ET were quantified.

It has to be said that these conclusions are not suprising, in fact are to be expected, and there are numerous studies published in hydrological literature that report the same or similar findings. In this sense, the potential innovation of the paper is rather limited, and would have to be found in the details and/or implications that are specific to the study cathments, climates, model used, etc. To highlight what is really innovative and make the relevance of the paper more clear I suggest to focus in the revision on the following points and questions.

1. What is the real aim of the paper? In the introduction (p2, 10-15) the authors claim that many studies have explored the effect of spatial resolution, but few (none) have explored the "effect of the modelling approach". It should be clarified from the start what is meant by this, because the authors in my opinion also only show the effect of spatial resolution and not modelling approach. They use the same conceptual model, same parameterisations, only the input data differ.

   The aim of this study is to quantify the differences in anomalies as result of a different spatial resolution, using a newly defined metric. We understand that our usage of "modelling approach" can be confusing. With modelling approach we link to the two scales of modelling: global and regional. In general, we identify two common practices related to the scale of modelling: a relatively coarse resolution for global studies, and relatively fine resolution for regional studies. In our study, we aim to compare these two "approaches" while keeping as many factors constant. We understand the confusion, and will clarify this in the revised version of the manuscript. Another innovation of our study is that, as far as we are aware, we are the first to show that it is not uncommon to have both extreme positive and negative flux anomalies occurring simultaneously within a catchment.

2. The studied catchments are all smaller than one coarse pixel in the analysis. This is quite clearly stated in Section 2.4. The input data are resampled to the 500x500 grid and averaged to a 40,000x40,000 m grid for the coarse application. So this sounds to me like comparing a spatially distributed model to a point model, not to a coarse resolution model. This also means that all elevation dependencies in hydrological processes in the point application are gone. Is this correct?

This is indeed correct, SPHY does only allow for sub-grid variability in glacier cover, but not in elevation via e.g. elevation zones. We are aware that some hydrological models do allow for different elevation zones within a single pixel, yet this still limits the model to output only a single value per pixel. In the "coarse resolution" model setup, each basin is captured within a single pixel, as would happen when simulating this region at 0.5x0.5° resolution. We chose for this configuration to present the difference in a "worst-case scenario" regarding model setup.

3. I would have liked to see at least a table with the values of the key parameters that were calibrated, i.e. something that gives more credibility to the SPHY model application. It seems in Section 2.4 that the model was calibrated for both the spatial application and the point application separately. I assume it is the high resolution application shown in Figure 5. How different were the parameter values? What do the differences (if any) mean for the results of the simulations, e.g. temperature dependencies, etc.

Thank you for this suggestion. We have added a table with parameter values in Table C1. In some cases, the optimization algorithm reach the parameter boundary. This happens more often with the low resolution model, indicating that SPHY struggled to simulate the hydrograph at these resolutions. Furthermore, we have also updated the calibration and validation figure, so it also includes the discharge simulations with the low resolution model (see Fig. C1).

**Table C1.** Parameter values resulting from optimization, for both the high and low resolution models.

|  |  | rootdepth | tcrit | snowsc | ddfsnow |
|---|---|---|---|---|---|
| high resolution | Reuss | 461.22 | -0.99 | 0.01 | 2.11 |
|  | Rhone | 1719.33 | 0.06 | 0.18 | 2.10 |
|  | Inn | 2997.40 | -0.86 | 0.01 | 1.38 |
|  | Emme | 453.35 | -1.00 | 0.99 | 3.68 |
|  | Thur | 402.60 | -0.14 | 0.03 | 1.05 |
| low resolution | Reuss | 402.60 | -0.75 | 0.01 | 1.01 |
|  | Rhone | 2997.40 | -0.40 | 0.01 | 1.01 |
|  | Inn | 2997.40 | -1.00 | 0.01 | 1.01 |
|  | Emme | 1919.75 | -0.40 | 0.70 | 3.38 |
|  | Thur | 537.97 | 0.24 | 0.19 | 5.41 |

4. The results were compared on seasonal anomalies of runoff and ET, summed over the catchment areas. There are no supporting data and plots to actually show how the model performed for spatially distributed variables, beyond Figure 6.

[Figure]

**Figure C1.** Calibration and validation of SHPY. The model is validated over the entire period, excluding the calibration period. The two black lines, and the shaded areas present the average discharge plus/minus the standard deviation. For Inn, all data after 2003 was excluded, since the observed discharge pattern changed after this period.

For example, snow cover could have easily been compared with data to show how snow accumulation and melt processes are simulated. There is little confidence given to the spatial predictions of the model, on which the entire analysis is based.

This was also stated by the first reviewer, and we added an internal validation of the model by comparing it with observations from the Rietholzbach, a research catchment situated in the Thur basin (Seneviratne et al., 2012). We obtained the evaporation data as measured with a long-term research lysimeter, and discharge data from this catchment. Both discharge and evaporation from the corresponding pixel were extracted from SPHY, to compare with the observations. We calculated the anomalies over the entire simulation period. The comparison between the observed and simulated anomalies can be found in Table C2. This table shows that the simulated anomalies agree well with the direction and magnitude of the observed anomalies. Winter and autumn values for evaporation are gray, since they are not the focus of this study due to the fact that SPHY does not allow for evaporation during snow covered periods. We believe that this table is a valuable addition to the validation of the model.

There is a slight mismatch between the evaporation anomalies during the summer of 2003, yet both describe unusually high values. To further investigate this, we also plotted the evaporation time series in Figure C2. This figure shows that

**Table C2.** Comparison between anomalies simulated with SPHY and observed in the Rietholzbach, anomalies are based of the entire simulation period.

| Event | Runoff | | Evaporation | |
|---|---|---|---|---|
| | Observed | Simulated | Observed | Simulated |
| DJF 1995 | 1.68 | 2.34 | 0.86 | 0.53 |
| MAM 2007 | -0.52 | -0.27 | 1.61 | 0.98 |
| JJA 2003 | -2.15 | -2.17 | 1.66 | 3.61 |
| SON 2002 | 2.65 | 2.62 | -1.76 | -0.26 |

SPHY is able to accurately simulate the evaporation, yet there are some differences between the two time series. These can be attributed to the scale difference between the lysimeter and a single high resolution SPHY pixel, and the fact that SPHY does not account for all factors influencing evaporation since it uses the temperature-based Hargreaves method. The difference between the two time series influences the mean and standard deviation, and therefore the resulting anomaly values.

[Figure]

**Figure C2.** Comparison between observed evaporation in a lysimeter, and the simulated evaporation of the corresponding pixel.

5. A new metric DWD is proposed to show the place of the point model value in the spatial distribution of the spatially resolved values. This is an interesting metric. I appreciate the effort to illustrate its use in Figure 3.

   Thank you for your kind words. Please note that we slightly adapted the equations of this metric, so that the distances are properly weighted. This new formulation only has a limited effect on the results, and does not alter the conclusions. See our detailed response to Davide Zoccatelli for the new equations and the effect on the DWD values.

10

6. The relation of the seasonal anomalies in ET and runoff to temperature across the basins in Figure 7 is interesting. I have one concern that the results here are probably strongly dependent on the parameterisations and structure of the model. Some of these relations, e.g. between runoff and temperature can be gleaned directly from observations. Will you get the same sensitivities? In Figure 8 the anomalies are plotted for every grid cell as a function of land cover. What about soil depth? Does SPHY assume that soil depth and soil properties are constant in space?

The simulations are indeed strongly dependent on the model structure and parameterization. However, since we analyzed all results on a three monthly time step, we do not expect the model to be a large factor here, but that differences in forcing are of great importance. SPHY does assume soil depth to be constant in space (the rootzone parameter in Table C1), but soil properties are related to the soil type and are variable in space. We chose to plot the anomalies against land cover since this corresponds to the studies by Theurillat and Guisan (2001) and Jolly et al. (2005). Since land cover and soil properties are related, we do not expect the results to be very different.

7. Overall, I find the physical relations between the results and hydrological processes in Section 3.1 nicely covered, the arguments are logical, especially the elevation effect is coming out strongly. I suggest also looking at the paper by Fatichi et al. (2015) "High resolution distributed analysis of climate and anthropogenic changes on the hydrology of an Alpine catchment" in Journal of Hydrology for another demonstraiton of this effect with a physically-based model.

Thank you for highlighting this paper. Really interesting to see the strong influence of elevation regardless of climate model used. We will cite this paper in our manuscript as it will be a valuable addition to put our result in perspective.

8. As mentioned above, the resolution effects are less instructive than the explanation of the anomalies. I am not sure what to take out of Figure 9, other than the point model lies within the range of the cells of the distributed model. This was for an extreme year, what about the entire simulation? Probably the results are the same. The message simply seems to be that lumping in space averages hydrological response, which is something very well known. Is there more to it than that? If yes, this has to be brought to the forefront more clearly

We have included Figure 9 to visualize the DWD results in Table 2. This figure helps to explain why we see some large DWD values in the table, especially for the ET results. During seasons with less or no extreme conditions, the violinplots from the high resolution model are generally less spread out (less variation in anomaly values), also resulting in lower DWD values. We agree that this is currently not clearly explained in the text, and will add this to the revised version of the manuscript.

**References**

Jolly, W. M., Dobbertin, M., Zimmermann, N. E., and Reichstein, M. (2005). Divergent vegetation growth responses to the 2003 heat wave in the Swiss Alps. *Geophysical Research Letters*, 32(18):L18409.

Seneviratne, S. I., Lehner, I., Gurtz, J., Teuling, A. J., Lang, H., Moser, U., Grebner, D., Menzel, L., Schroff, K., Vitvar, T., and Zappa, M. (2012). Swiss prealpine Rietholzbach research catchment and lysimeter: 32 year time series and 2003 drought event. *Water Resources Research*, 48(6):W06526.

Theurillat, J.-P. and Guisan, A. (2001). Potential Impact of Climate Change on Vegetation in the European Alps: A Review. *Climatic Change*, 50(1-2):77–109.

---

## Author Response (AR1)

**Author's response**

1. We have revised the abstract and introduction to better state the aim of our study, and to better embed it in existing literature.

2. We have added average annual precipitation to Table 1.

3. We have updated the DWD equations according to the comments made by Davide Zoccatelli, and updated the relevant figures and tables.

4. The calibration and validation figure was updated to include results from both models, and added a better visualization of the validation of the model. We have also added a comparison with observations from the Rietholzbach research catchment to provide an internal model validation.

5. We have added a figure to compare the aggregated high resolution model response with the low resolution model response, as was suggested by Davide Zoccatelli.

[revised manuscript text omitted]

$$\mathrm{DWD} = (P_{\mathrm{low\_res}} - 0.05) \cdot d_{\mathrm{lower}} + (0.95 - P_{\mathrm{low\_res}}) \cdot d_{\mathrm{upper}},$$

10

$$DWD = W_{\mathrm{lower}} \cdot d_{\mathrm{lower}} + W_{\mathrm{lower}} \cdot d_{\mathrm{lower}}, \tag{3}$$

$$W_{\mathrm{lower}} = \max\left(0, \min\left(1, \frac{P_{\mathrm{low\_res}} - P_{\mathrm{lower}}}{P_{\mathrm{upper}} - P_{\mathrm{lower}}}\right)\right), \tag{4}$$

$$W_{\mathrm{upper}} = \max\left(0, \min\left(1, \frac{P_{\mathrm{upper}} - P_{\mathrm{low\_res}}}{P_{\mathrm{upper}} - P_{\mathrm{lower}}}\right)\right), \tag{5}$$

$$d_{\mathrm{lower}} = Z_{\mathrm{low\_res}} - Z^{5\%}_{\mathrm{high\_res}}, \tag{6}$$

15 $$d_{\mathrm{upper}} = Z^{95\%}_{\mathrm{high\_res}} - Z_{\mathrm{low\_res}}, \tag{7}$$

where $W_{\mathrm{lower}}$ and $W_{\mathrm{upper}}$ are the weights used to weigh the distances $d_{\mathrm{lower}}$ and $d_{\mathrm{upper}}$  . $P_{\mathrm{low\_res}}$ is the percentile of $Z_{\mathrm{low\_res}}$ within $Z_{\mathrm{
[revised manuscript text omitted]

---

## Author Response (AR2)

**Author's response**

1. Based on the suggestion made by Davide Zoccatelli, we now also describe how our results compare with previous studies, and how our manuscript fits within the existing literature on this subject.

2. We have rephrased parts of the conclusion to better represent what the DWD metric quantifies, and to avoid confusion.

5   3. We have moved Appendix A to supplementary information.

4. We have incorporated the other improvement points suggested the referee and the editor.

[revised manuscript text omitted]